# Association of Bitter Taste Receptors with Obesity and Diabetes and Their Role in Related Tissues

**Eisuke Kato** [1,*] and **Shota Oshima** [2]

1    Division of Fundamental AgriScience and Research, Research Faculty of Agriculture, Hokkaido University, Kita-ku, Sapporo 060-8589, Japan
2    Frontiers in Bioscience, Graduate School of Agriculture, Hokkaido University, Kita-ku, Sapporo 060-8589, Japan
*    Correspondence: eikato@agr.hokudai.ac.jp

**Abstract:** Taste 2 receptors (T2Rs) are G-protein-coupled receptors responsible for sensing bitter tastes. Many studies have shown the expression of T2Rs in extraoral tissues and the unique role of T2Rs in each tissue. Single-nucleotide polymorphisms of T2Rs are associated with the risk of obesity and diabetes, and the organs/tissues associated with the development of these metabolic diseases, including the intestine, adipose, muscle, liver, and pancreas, are reported to express T2R genes. This result suggests that T2Rs in extraoral tissues contribute to the development of obesity and diabetes. In this narrative review, we summarize current knowledge of the associations of T2Rs with obesity and diabetes, provide an overview of extraoral tissues that are associated with the development of obesity and diabetes that express T2R genes, and summarize the current knowledge of T2Rs.

**Keywords:** taste 2 receptor; bitter taste receptor; obesity; diabetes

## 1. Introduction

Vertebrates sense bitterness through bitter taste receptors named taste 2 receptors (T2Rs). T2Rs are characterized by their genetic diversity compared with other taste receptors. For example, humans and rodents have approximately 25 and 36 functional T2R genes, respectively [1,2]. Because bitterness is an unpleasant taste and bitter compounds are generally considered to have undesirable effects on health, this diversity is thought to allow for the perception of a variety of bitter compounds and avoidance of their intake.

T2Rs are expressed in taste receptor cells located in taste buds on the tongue and soft palate. However, T2R expression is not limited to the oral cavity; many studies have shown that T2R genes are expressed in a variety of organs/tissues/cells, including the respiratory tract, vascular system, brain, stomach, testis, muscle, and fat cells [3]. These extraoral T2Rs that are not involved in bitter taste perception have roles that vary from tissue to tissue.

In many tissues, extraoral T2Rs are involved in protecting the body from harmful agents. This has been well studied for tissues that come into contact with viruses, microorganisms, and parasites. In the upper respiratory tract, T2Rs are expressed on solitary chemosensory and ciliated cells, where they sense pathogens and induce a defense response [4]. Meanwhile, some studies have suggested that the roles of T2Rs are not limited to biological defense; one such suggested role is the regulation of lipid/glucose metabolism, as reports have shown correlations of T2Rs with obesity and diabetes [5–9].

This narrative review focuses on the association of T2Rs with obesity and diabetes. First, genetic studies suggesting such a relationship are summarized, followed by a discussion of the expression and function of T2Rs in tissues important for glucose and lipid metabolism. To clarify the relationship between bitter compounds and T2Rs described in the text, the agonists and their known target T2Rs are summarized (Table 1). Expression of T2R in extraoral tissues and the studies suggesting the role of T2R are summarized in Tables 2–4.

**Table 1.** Bitter compounds and their reported target T2Rs.

| Compound | Target (Human) | Ref. | Target (Mouse) | Ref. |
|---|---|---|---|---|
| allyl isothiocyanate | TAS2R38 | [10] | Tas2r135 | [11] |
| caffeine | TAS2R7, 10, 14, 43, 46 | [10] | Tas2r121 | [11] |
| chloroquine | TAS2R3, 7, 10, 39 | [10] | Tas2r115 | [11] |
| denatonium benzoate | TAS2R4, 8, 10, 13, 39, 43, 46, 47 | [10] | Tas2r105, 123. 135, 140, 144 | [11] |
| dextromethorphan | TAS2R1, 10 | [10] | | |
| epicatechin | TAS2R4, 5, 39 | [12] | Tas2r126, 144 | [11] |
| epigallocatechin-3-gallate | TAS2R14, 39 | [13] | Tas2r144 | [11] |
| isorhamnetin | TAS2R14, 39 | [13] | | |
| KDT501 | TAS2R1 | [14] | Tas2r108 | [14] |
| luteolin | TAS2R14, 39 | [13] | | |
| nobiletin | TAS2R14 | [15] | | |
| noscapine | TAS2R14 | [10] | | |
| oleuropein | TAS2R8 | [16] | | |
| 6-propyl-2-thiouracil (Prop) | TAS2R4, 38 | [10] | Tas2r105, 108, 120, 121, 135, 137 | [11] |
| quercetin | TAS2R14 | [13] | | |
| quinine | TAS2R4, 7, 10, 14, 39, 40, 43, 44, 46 | [10] | Tas2r105, 108, 115, 126. 137, 140, 144 | [11] |
| resveratrol | TAS2R14, 39 | [13] | Tas2r108, 109, 131, and 137 | [17] |
| saccharin | TAS2R8, 43, 44 | [10] | Tas2r105, 109, 135, 144 | [11] |
| salicylic acid | | | Tas2r135 | [11] |
| silibinin | TAS2R14, 39 | [13] | | |

**Table 2.** Expression of T2R in rodents.

| Tissue or Cell | Expression | Ref. |
|---|---|---|
| Intestine | *Tas2r108, 126, 135, 137, 138*, and *143* | [18,19] |
| Inguinal WAT | *Tas2r108, 113, 118, 119, 126, 135, 137, 138, 140, 143*, and *144* | [19] |
| Hind limb skeletal muscle | *Tas2r108, 126, 134, 135, 137, 140, 143*, and *144* | [19] |
| Vascular smooth muscle | *Tas2r116* and *Tas2r143* | [20] |
| Liver | *Tas2r108, 126, 135, 137, 138*, and *143* | [18,21] |
| Liver | *Tas2r108, 109, 126, 130, 135, 137, 138*, and *143* | [19] |

**Table 3.** Expression of T2R in humans.

| Tissue or Cell | Expression | Ref. |
|---|---|---|
| Intestine | *TAS2R4, 5, 14, 20* (High), *TAS2R3, 10, 13, 19, 30, 31, 38, 43, 46, 50*, and *60* (Low) | GTEx |
| Intestinal L-cells | TAS2R38 | [22] |
| Subcutaneous WAT | *TAS2R14, 19, 45*, and *46* (High) *TAS2R3, 7, 31*, and *43* (intermediate), *TAS2R5, 10, 13, 20*, and *39* (Low) | [19] |
| Subcutaneous and visceral WAT | *TAS2R5, 14*, and *20* (High), *TAS2R4, 10, 19*, and *31* (intermediate), *TAS2R3, 13, 43, 46*, and *50* (Low) | GTEx |
| Airway smooth muscle | *TAS2R1, 3, 4, 5, 8, 9, 10, 13, 14, 19, 30, 31, 42, 45, 46*, and *50* | [23] |
| Vascular smooth muscle | *TAS2R3, 4, 7, 10, 14, 39*, and *40* | [24] |
| | TAS2R46 | [20] |
| Cardiac muscle | *TAS2R3, 4, 5, 9, 10, 13, 14, 19, 20, 30, 31, 39, 43, 45, 46*, and *50* | [20] |
| Gastrocnemius muscle | *TAS2R5, 14*, and *20* (High), *TAS2R4* and *19* (intermediate), *TAS2R3, 10, 13, 30, 31, 43*, and *50* (Low) | GTEx |
| Liver | *TAS2R4, 5, 10, 13, 14, 19, 20, 30, 31, 43*, and *46* | GTEx |
| Islets of Langerhans | *TAS2R3, 4, 5, 9, 10, 13, 14, 19, 31, 43, 45, 46, 50*, and *60* | [25] |
| Pancreas | *TAS2R3, 4, 5, 10, 14, 19, 20*, and *31* | GTEx |

**Table 4.** Studies suggesting the role of T2R in tissues is important for glucose and lipid metabolism.

| Tissue | Experimental Model | Specie | Summary | Ref. |
|---|---|---|---|---|
| Intestine | NCI-H716 cells isolated proximal duodenum | human mouse | Denatonium and quinine stimulate cells to secrete GLP-1, and the knockdown of *TAS2R3, 44,* and *46* decreased the response. Isolated proximal duodenum secretes GLP-1 in response to denatonium. | [26] |
| | HuTu-80 cells BALB/c mice | human mouse | PROP and Z7 stimulate cells to secrete GLP-1, and the knockdown of *TAS2R38* decreased the response. Oral administration of TAS2R38 ligand increased serum GLP-1 levels. | [22] |
| | NCI-H716cells STC-1 cells | human | Berberine upregulates the secretion of GLP-1, and the knockdown of *TAS2R38* decreased the response. | [27,28] |
| | diet-induced obese mice | mouse | Oral gavage of KDT501 increased plasma GLP-1 levels | [14] |
| | healthy men | human | intraduodenal administration of quinine increased plasma GLP-1 levels. | [29] |
| Adipose | primary preadipocytes | mouse | Quinine stimulates adipogenesis, and *Tas2r106* knockdown suppressed the action. | [30] |
| | 3T3-L1 preadipocytes | mouse | Overexpression of *Tas2r108* or *Tas2r126* reduced adipogenesis | [31] |
| | primary adipocytes | human | PROP, quinine, and caffeine reduce lipid accumulation and increase the expression of *TAS2R38* | [32] |
| | 3T3-F442A adipocytes | mouse | Denatonium benzoate and quinine reduce lipid accumulation and increase the expression of *Tas2r108* and *Tas2r135.* | [33] |
| | 3T3-L1 adipocytes | mouse | Caffeine reduces lipid accumulation | [34,35] |
| | 3T3-L1 adipocytes | mouse | Nobiletin, isorhamnetin, and salicylic acid upregulate brown adipocyte marker gene or protein. | [36–38] |
| | diet-induced obese mice | mouse | Oral gavage of epicatechin increases BAT-specific markers in perivisceral subcutaneous adipose tissue. | [39] |
| | high-fat diet-fed mice | mouse | Supplementation of luteolin in diet promoted thermogenesis in BAT and subcutaneous adipose tissue. | [40] |
| | primary adipocytes | human | Silibinin increases thermogenic marker genes | [41] |
| | primary adipocytes | human | Caffeine increased UCP-1 level | [42] |
| Muscle | isolated trachea | mouse | Chloroquine, quinine, and denatonium benzoate induce relaxation | [23] |
| | airway smooth muscle cells | human | Saccharine and chloroquine increase intracellular $Ca^{2+}$ | [23] |
| | isolated bronchial smooth muscle | rat and mouse | Denatonium or PROP induces relaxation | [43] |
| | isolated aorta ring | guinea pig | Chloroquine, denatonium, dextromethorphan, or noscapine induce relaxation | [44] |
| | isolated pulmonary arteries | human | Chloroquine, dextromethorphan, or noscapine induce relaxation | [44] |
| | vascular smooth muscle cells | human and rat | Denatonium increases intracellular $Ca^{2+}$ | [20] |
| | isolated ileal smooth muscle | mouse | Responds to denatonium or PROP | [43] |
| | isolated abdominal skeletal muscle | rat | Denatonium induce relaxation | [45] |
| | gastric smooth muscle cells | human | Denatonium benzoate induces contraction and relaxation at different concentration | [46] |
| Liver | high-fat diet-fed mice | mouse | Supplementation of oleuropein to diet reduced liver weight and hepatic triglyceride level | [47] |
| | diet-induced obese mice | mouse | Oral gavage of KDT501 reduced lipid deposition of the liver | [48] |
| | high-fat diet-fed mice | mouse | Supplementation of epigallocatechin-3-gallate to diet reduced hepatic lipid and cholesterol content | [49] |
| Islet | HIT-T15 cells isolated islet | hamster rat | Denatonium benzoate induces insulin secretion | [50] |
| | isolated islet | rat | β-L-Glucose pentaacetate induces insulin secretion | [51] |

## 2. Relationship of T2R, Obesity, and Diabetes

Several studies have suggested that T2Rs are involved in the regulation of body fat mass in humans. TAS2R38 is a bitter taste receptor that has commonly been the focus of studies of genetic variation that distinguishes those able and unable to taste 6-propyl-2-thiouracil (PROP). The genotyping of women with anorexia nervosa, healthy controls, and morbidly obese patients, or of three ethnically diverse groups (European Americans, African Americans, and Asians) has suggested an association between single-nucleotide polymorphism in *TAS2R38* and the development of obesity [5,6]. Single-nucleotide polymorphisms at three positions in *TAS2R38* alter amino acids and produce two haplotypes: PAV (proline-alanine-valine) and AVI (alanine-valine-isoleucine). Of these two haplotypes, obesity was found to be more common in those with AVI (those unable to taste PROP) than in those with PAV (those able to taste PROP). In another study, minor alleles of polymorphisms in *TAS2R4* and *TAS2R5*, the receptors that detect the dietary polyphenol epicatechin [12], were found to be associated with lower BMI [7].

The manipulation of signal transduction pathways typically linked to T2R activation influences the progression of obesity. Knockout of α-gustducin in C57BL/6 mice was found to increase thermogenesis and protect against high-fat-diet-induced obesity despite increased energy intake [33]. Loss of α-gustducin diminishes the signal from T2Rs (see Figure 2 for the signaling pathway). It is thus indicated that T2R signaling leads to increased body fat mass upon consumption of a high-fat diet. The increased heat production was explained by the increased expression of UCP-1, which was confirmed in white adipose tissue (WAT) at the mRNA expression level and brown adipose tissue (BAT) at the protein level, suggesting a role of T2Rs in these tissues [33]. However, α-gustducin participates in the signaling of taste 1 receptors (T1Rs), which sense sweet and umami tastes, and loss of α-gustducin also reduces signaling from the T1Rs. Adipose tissue has also been shown to express T1Rs [52], and the differences observed in α-gustducin-knockout mice should be considered to include effects owing to the loss of signaling through T1Rs.

The association between single-nucleotide polymorphisms in the human T2R genes and the risk of type 2 diabetes has been reported by Dotson et al. [8]. Through the geno-typing of an Amish family and the oral glucose tolerance test, they found that two single-nucleotide polymorphisms in TAS2R7, one in a non-coding region and the other in a coding region, and one single-nucleotide polymorphism in TAS2R9 in a coding region, are associated with the regulation of glucose and insulin levels. They also showed that the single-nucleotide polymorphism in TAS2R9 alters the amino acid sequence (Ala187 to Val187), leading to a diminished response upon stimulation by ofloxacin, procainamide, and pirenzepine in cellular models. These results led to the conclusion that polymorphism in T2Rs alters the response to their ligands, and this altered response may have an influence on glucose and insulin homeostasis.

## 3. T2Rs in Tissues Associated with the Development of Obesity and Diabetes

### 3.1. Intestine

#### 3.1.1. Expression of T2R Genes in Intestine

The intestine absorbs nutrients as well as secretion of incretin hormones to participate in the development of diabetes and obesity. Several groups have reported the expression of T2Rs in gut tissue. For example, whole-mouse tissue analysis by Prandi et al. showed consistent expression of *Tas2r108*, *126*, *135*, *137*, *138*, and *143* throughout the gut (stomach, small, and large intestine) together with several T2R genes expressed in specific organs [18]. Our group also found the expression of *Tas2r108*, *126*, *135*, *137*, *138*, and *143* in the small intestine of C57BL/6J mice [19]. More precisely, tuft cells [53], Paneth cells [18], goblet cells [18], and enteroendocrine cells [26,54] in the gut tissue of mice were found to express T2R genes.

Vagezzi et al. showed that the expression of *Tas2r138* in the gut is regulated by diet or diet-associated changes in intraluminal conditions [55]. Specifically, they obtained the following findings: (1) fasting decreases and re-feeding restores the expression of *Tas2r138* in the stomach; (2) feeding on a diet supplemented with lovastatin and ezetimibe to deplete cholesterol absorption for 1 week upregulated the expression of *Tas2r138* in the duodenum, jejunum, and proximal colon; and (3) feeding on a high-fat diet for 8 weeks, but not 2 weeks, upregulated the expression of *Tas2r138* in the colon. The exact mechanism by which the expression of *Tas2r138* is regulated was not confirmed, but Vagezzi et al. suggested the following: (1) fasting-induced decrease in *Tas2r138* in the stomach might be regulated by the taste-related molecules contained in the gastrin and ghrelin cells; (2) cholesterol depletion enhances cholesterol-sensitive SREBP-2 expression and enhances the expression of *Tas2r138* [56]; and (3) the unaltered expression of *Tas2r138* by short-term feeding on a high-fat diet suggests that a high-fat diet does not directly influence such expression, and long-term feeding on a high-fat diet to alter the intraluminal conditions, especially the gut microbiota, might enhance such expression to serve as a defensive mechanism against pathogenic bacteria. Although the assumed mechanisms regulating the expression

of *Tas2r138* differ in each condition, these results suggest an association of the diet or diet-associated changes with the expression of *Tas2r138*.

In humans, as shown in the Genotype-Tissue Expression (GTEx) database, the expression of *TAS2R4, 5, 14,* and *20* is relatively high in the small intestine and colon, whereas the expression of *TAS2R3, 10, 13, 19, 30, 31, 38, 43, 46, 50,* and *60* is moderate to low (Figure 1). Moreover, Pham et al. showed that human enteroendocrine L-cells isolated from ileum tissues express TAS2R38 protein [22].

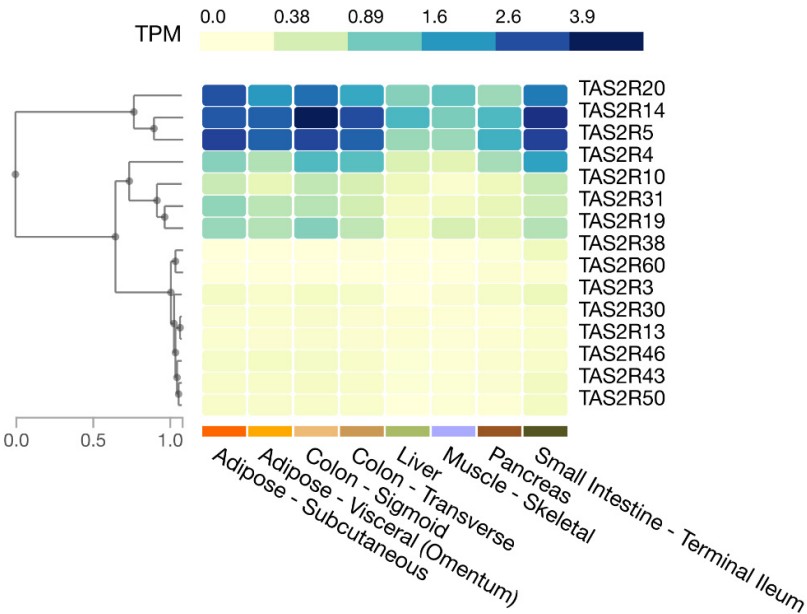

**Figure 1.** Expression of T2R genes in selected human tissues. Figure created in the GTEx portal (https://gtexportal.org/home/, accessed on 21 July 2023).

3.1.2. Role of T2Rs in Intestine

T2Rs in intestinal tuft cells activate immune responses in response to parasites and their secretory products in mice [53]. The role of T2Rs in Paneth cells and goblet cells is unclear, but it has been suggested that they are involved in protection against pathogens through the regulation of antimicrobial responses and the production of mucins [18]. Therefore, T2Rs in these types of cells are unlikely to be involved in the development of obesity or diabetes.

Some T2Rs are expressed in enteroendocrine cells, and therefore, they are likely to be involved in obesity and diabetes. T2Rs in enteroendocrine cells regulate the secretion of the hormone incretin, which in turn promotes the secretion of insulin from the pancreas. The human enteroendocrine cells NCI-H716 secrete glucagon-like peptide-1 (GLP-1) in response to stimulation with denatonium benzoate and quinine, but knockdown of *TAS2R3, 44,* and *46* results in decreased secretion [26]. That study also showed that the mouse proximal duodenum secretes GLP-1 in response to denatonium benzoate. Cells of the human duodenum adenocarcinoma HuTu-80 have been shown to secrete GLP-1 in response to PROP and *Z7* (the ligand of TAS2R38), but such secretion is reduced when *TAS2R38* is knocked down [22]. In that study, Pham et al. also showed that the oral administration of TAS2R38 ligand increased serum GLP-1 levels in BALB/c mice. In other studies, Yu et al. and Yue et al. showed that berberine upregulates the secretion of GLP-1 from NCI-H716 and STC-1 enteroendocrine cells [27,28]. Upregulation of GLP-1 secretion was inhibited by the knockdown of *TAS2R38* in both studies, indicating the contribution of T2Rs to this response.

In a mouse study, Kok et al. examined the effect of KDT501, a derivative of the bitter compound isohumulone contained in hops, in mice with diet-induced obesity [14]. The oral administration of a single dose of KDT501 to fasted mice increased plasma GLP-1 levels,

which also improved the glucose clearance in the oral glucose tolerance test. The intragastric administration of KDT501 for 28 days improved glucose homeostasis parameters and led to reduced body weight due to a reduction in fat mass. Meanwhile, a human study by Rose et al. showed that the intraduodenal administration of quinine increased plasma GLP-1 levels, while the consumption of a mixed-nutrient drink led to a reduction in the elevation of plasma glucose level [29].

These reports show that T2R activation has the potential to stimulate GLP-1 secretion from intestinal enteroendocrine cells and contribute to glycemic control.

### 3.2. Adipose Tissue

### 3.2.1. Expression of T2R Genes in Adipose Tissue

Adipose tissue accumulates lipids and is directly related to obesity, and its insulin sensitivity is associated with the development of diabetes. Amisten et al. show that human subcutaneous adipose tissue expresses relatively high levels of *TAS2R14*, *19*, *45*, and *46*; intermediate levels of *TAS2R3*, *7*, *31*, and *43*; and low or trace levels of *TAS2R5*, *10*, *13*, *20*, and *39* [57]. The GTEx database shows relatively high levels of *TAS2R5*, *14*, and *20*; intermediate levels of *TAS2R4*, *10*, *19*, and *31*; and low levels of *TAS2R3*, *13*, *43*, *46*, and *50* in human subcutaneous and visceral adipose tissue (Figure 1). The differences between the above report and the database may reflect differences in methods (qPCR vs. RNA sequencing) but may also involve individual differences because individual differences in T2R expression have been observed in mouse adipose tissues [19].

Our research group has focused on the expression of T2R genes in mouse adipose tissue [19]. In mouse inguinal white adipose tissue, the expression of *Tas2r108*, *113*, *118*, *119*, *126*, *135*, *137*, *138*, *140*, *143*, and *144* was detected, with *Tas2r108*, *126*, *135*, *137*, and *143* showing relatively high expression compared with the other genes. Expression of the latter five T2R genes was also observed in 3T3-L1 adipocytes, which are widely used as model cells of adipocytes [19].

### 3.2.2. Role of T2R in Maturation of Preadipocytes

Two studies have directly demonstrated the role of T2R in adipocyte differentiation [30,31]. A first study by Ning et al. showed that the addition of quinine to the differentiation medium of primary mouse preadipocytes enhanced lipid accumulation in differentiated adipocytes [30]. They also showed that adipogenic markers (*Pparg*, *Cebpa*, *Fabp4*) were elevated and that adipogenesis was stimulated by quinine. Moreover, they showed that the expression of *Tas2r106* in the adipocytes was increased during differentiation into mature adipocytes. The contribution of *Tas2r106* was shown by shRNA-induced knockdown, which suppressed quinine-mediated adipogenesis, indicating that Tas2r106 mediates the action of quinine [30].

A second study by our group showed that the expression of *Tas2r108*, *126*, *135*, *137*, and *143*, the five T2R genes primarily expressed in 3T3-L1 cells, increased two to three-fold during differentiation, suggesting their role in this process 31]. In support of this hypothesis, the treatment of 3T3-L1 cells with epicatechin, a bitter agonist of Tas2R126, led to the upregulation of transcription factors associated with adipocyte differentiation. The overexpression of *Tas2r108* or *Tas2r126* reduced lipid accumulation during differentiation and also reduced the expression of the adipocyte marker genes *Pparg* and *Cebpa*, indicating the inhibition of differentiation into mature adipocytes.

These two studies showed that T2Rs regulate the differentiation process. However, there was also a discrepancy in the obtained findings: the addition of quinine, the input signal from Tas2r106, promoted differentiation, whereas the overexpression of *Tas2r108* or *Tas2r126*, which are expected to increase signaling from T2Rs, suppressed differentiation. These results may be due to differences between *Tas2r106* and *Tas2r108/126* or between primary mouse adipocytes and 3T3-L1 adipocytes.

### 3.2.3. Role of T2Rs in Mature Adipocytes

Two studies have suggested that bitter compounds, and potentially T2Rs, have an effect on adipocytes. Cancello et al. showed that treatment of human-derived primary adipocytes with PROP, quinine, and caffeine reduced lipid accumulation with increased expression of *TAS2R38* [32]. In addition, Avau et al. reported that denatonium benzoate and quinine reduced lipid accumulation in mouse-derived 3T3-F442A adipocytes by modulating lipid metabolism and that these cells expressed *Tas2r108* and *Tas2r135*, the targets of the two compounds [33].

More studies have shown the effect of bitter compounds on adipocytes, although T2R involvement has not been suggested. In the culture of 3T3-L1 adipocytes, stimulation by caffeine reduced the lipid accumulation via the inhibition of insulin-stimulated glucose uptake and/or lipolytic activity [34,35], while nobiletin upregulated brown adipocyte marker proteins (PGC-1$\alpha$ and UCP-1) and fatty acid oxidation genes, accompanied by the phosphorylation of PKA and AMPK [36]. In another study, isorhamnetin reduced lipid content, increased the expression of PGC-1$\alpha$, and increased mitochondrial DNA, accompanied by increased AMPK activity [37], while salicylic acid activated AMPK, leading to an increase in the expression of PGC-1$\alpha$ and an increase in mitochondrial DNA [38]. In C57BL/6 mice, daily oral gavage of epicatechin increased the levels of mitochondrial biogenesis-related proteins and brown adipose tissue-specific marker protein in perivisceral subcutaneous adipose tissue [39], while another study showed that the dietary supplementation of luteolin upregulated thermogenic genes in brown and subcutaneous adipose tissues, together with enhancement of the expression of PGC-1$\alpha$ and phosphorylation of AMPK [40]. In human adipose stem cell-derived adipocytes, silibinin increased the gene expression of thermogenic markers and reduced the lipid content [41]. Additionally, in human stem cell-derived adipocytes, caffeine increased the UCP-1 level, and in a human study, the intake of caffeine resulted in an increased body temperature at the supraclavicular region where brown fat is located, which was assumed to result from the effect of caffeine on increasing BAT activity [42]. These studies show a common character of bitter compounds inducing the browning of adipocytes to enhance mitochondrial biogenesis and fatty acid oxidation.

These reports support the idea that T2Rs play some role in regulating lipid metabolism in mature adipocytes. Studies demonstrating a direct contribution of T2Rs to the effects of bitter compounds are warranted.

### *3.3. Muscle*

### 3.3.1. Expression of T2R Genes in Muscle

Muscle is an important tissue for postprandial glucose homeostasis. It also accounts for approximately 20–30% of resting energy expenditure. Postprandial hyperglycemia is a hallmark of diabetes, and obesity develops due to an imbalance between energy intake and expenditure. Thus, muscle is associated with the development of obesity and diabetes. Muscle tissues have been examined for the expression of T2R genes. It was found that human airway smooth muscle cells express *TAS2R1*, *3*, *4*, *5*, *8*, *9*, *10*, *13*, *14*, *19*, *30*, *31*, *42*, *45*, *46*, and *50*, with *TAS2R10*, *14*, and *31* being expressed at relatively high levels [23]. Human vascular smooth muscle cells express *TAS2R3*, *4*, *7*, *10*, *14*, *39*, and *40* [24], and the presence of TAS2R46 protein has also been reported [20]. Human cardiac muscle expresses *TAS2R3*, *4*, *5*, *9*, *10*, *13*, *14*, *19*, *20*, *30*, *31*, *39*, *43*, *45*, *46*, and *50* at higher levels than it expresses AGTR1 (angiotensin II type 1a receptor gene) [58]. In addition, according to the GTEx database, the expression of *TAS2R5*, *14*, and *20* is relatively high in skeletal muscle, while *TAS2R4* and *19* are expressed at intermediate levels, and *TAS2R3*, *10*, *13*, *30*, *31*, *43*, and *50* at low levels (Figure 1).

In mouse hind limb skeletal muscle, the expression levels of *Tas2r108*, *126*, *134*, *135*, *137*, *140*, *143*, and *144* were observed, while the skeletal muscle model cells C2C12 were also found to express *Tas2r108*, *126*, *135*, *137*, and *143* [19]. Moreover, rat vascular smooth muscle cells were also shown to express *Tas2r116* and *Tas2r143* [20].

3.3.2. Role of T2Rs in Muscle Tissue

Currently, there is no direct evidence for the role of T2Rs in muscle cells. However, several reports have shown the effects of bitter compounds on muscle tissues or cells.

The contribution of T2Rs to the muscle relaxation process was first demonstrated in airway smooth muscle, with Deshpande et al. showing the relaxation of mouse trachea induced by T2R agonists such as chloroquine, quinine, and denatonium benzoate [23]. Chloroquine-induced relaxation was found to be the result of the direct action of bitter compounds on mouse airway smooth muscle cells and was shown to work alongside the effect of β-adrenergic agonists in an additive manner, with no increase in intracellular cAMP levels but an increase in intracellular calcium ion concentration [23]. It was also shown that human airway smooth muscle cells respond to saccharine and chloroquine to increase intracellular calcium ion concentrations [23]. Similar results were shown by Sakai et al., who found that relaxation of rat and mouse bronchial smooth muscle was induced by denatonium or PROP and worked alongside the effect of β-adrenergic agonists in an additive manner [43].

Vascular smooth muscle cells also respond to T2R agonists. One study found that the treatment of endothelium-denuded guinea pig aortic ring with chloroquine, denatonium, dextromethorphan, or noscapine [44]; human pulmonary arteries with chloroquine, dextromethorphan, or noscapine [44]; and mouse aortic smooth muscle with denatonium or PROP [43] induced relaxation in all cases. Denatonium was also shown to increase intracellular calcium concentrations in human and rat vascular smooth muscle cells [20].

In addition to the two muscles mentioned above, ileal smooth muscle was shown to respond to denatonium or PROP [43], abdominal skeletal muscle was found to respond to denatonium and induce relaxation [45], and gastrointestinal muscle was identified to contract at low concentrations and relax at high concentrations in response to denatonium benzoate [46].

The expression of T2R genes in muscle tissues/cells and the relaxation/contraction effects of T2R agonists suggest the involvement of T2Rs. However, of the T2R agonists listed, denatonium and quinine were shown to act as α1-adrenergic receptor antagonists, leading to muscle relaxation [44]. The mechanisms associated with other T2R agonists remain to be elucidated, leaving open the possibility of a role of T2Rs in muscle relaxation.

All of the above findings link T2Rs to muscle movement, whereas no link between muscle T2Rs and obesity or diabetes has yet been revealed.

*3.4. Liver*

The liver is an organ that controls blood glucose levels by storing excess glucose as glycogen and generating glucose through glycogenesis. Obesity leads to the dysregulation of liver function through the development of non-alcoholic fatty liver disease. Thus, the liver is important in relation to diabetes and obesity.

The expression of *Tas2r108*, *126*, *135*, *137*, *138*, and *143* in mouse liver was reported by Prandi et al. [18] and Kurtz et al. [21]. Our group further found the expression of *Tas2r109* and *130* in mouse liver [19]. Meanwhile, the GTEx database shows that *TAS2R4*, *5*, *10*, *13*, *14*, *19*, *20*, *30*, *31*, *43*, and *46* are expressed in the human liver (Figure 1).

To date, no reports have been published suggesting a role of T2Rs in hepatocytes. However, several T2R agonists have been reported to reduce lipid levels in the liver of mice with high-fat-diet-induced obesity. In C57BL/6N mice, the supplementation of oleuropein, a constituent of olives, to a high-fat diet was found to reduce liver weight and hepatic triglyceride level, along with reduced expression of genes related to lipid metabolism [47]. The daily oral administration of KDT501 also reduced lipid deposition in the liver in mice with high-fat-diet-induced obesity [48]. Moreover, supplementation of epigallocatechin-3-gallate to a high-fat western-style diet reduced hepatic lipid and cholesterol content with altered hepatic bile acid and cholesterol metabolism, although decreased intestinal bile acid reabsorption and decreased lipid absorption were considered to participate in the

effect [49]. In vivo studies have also described other tissue-mediated effects, but hepatocyte T2Rs may contribute to these effects to some extent.

### 3.5. Pancreatic Islets

Pancreatic islets contain cells that secrete hormones regulating glucose metabolism. α-Cells secrete glucagon, which acts to increase blood glucose levels by stimulating glycogen hydrolysis in the liver. β-Cells secrete insulin, which decreases gluconeogenesis in the liver and increases glucose uptake by the liver, muscle, and adipose tissue, thereby lowering blood glucose levels. Thus, the pancreas is important in the pathogenesis of diabetes.

An analysis of T2R gene expression in human islets of Langerhans revealed the expression of *TAS2R3*, *4*, *5*, *9*, *10*, *13*, *14*, *19*, *31*, *43*, *45*, *46*, *50*, and *60* [25]. The GTEx database also shows that *TAS2R3*, *4*, *5*, *10*, *14*, *19*, *20*, and *31* are expressed in the pancreas (Figure 1); however, the role of T2Rs in the pancreas has not been carefully studied. Nonetheless, some bitter compounds have been reported to induce insulin secretion in isolated pancreatic islets, including denatonium benzoate and β-L-glucose pentaacetate [50,51]. One study showed that the effect of denatonium benzoate on insulin secretion was apparently mediated by a transducin-independent pathway, involving a decrease in ATP-sensitive potassium (KATP) channel activity, depolarization of β-cells, and increased $Ca^{2+}$ influx [50], casting doubt on the involvement of T2Rs. Notably, the activation of G-proteins other than transducin or gustducin by T2Rs has been reported [53,59–62] (see Section 4 for more detail), and thus, further investigation is necessary.

## 4. Signaling Pathway of Taste 2 Receptors in Extraoral Tissues

In taste cells, T2Rs interact with the heterotrimeric G-protein composed of α-gustducin, beta-3, and gamma-13 subunits. When activated by ligands, the beta and gamma subunits are released and activate phospholipase C β2 to synthesize inositol 1,4,5-triphosphate (IP3). IP3 then activates IP3 receptors on the surface of the endoplasmic reticulum, releasing $Ca^{2+}$ and increasing the intracellular $Ca^{2+}$ concentration. The increased $Ca^{2+}$ concentration leads to activation of the transient receptor potential cation channel, subfamily M, member 5 (TRPM5), which depolarizes the plasma membrane. Finally, calcium homeostasis modulator 1/3 (CALHM1/3) is activated, releasing ATP to transmit bitter taste signals to the taste nerve (Figure 2a) [63].

Among the above pathways, the co-presence of α-gustducin, in particular, was employed as evidence for the existence of functional T2Rs in extraoral cells. Intestinal tuft cells or enteroendocrine STC-1 cells fit this criterion, expressing α-gustducin together with T2Rs [27,53]. However, neither the presence of α-gustducin nor the expression of genes encoding it has been confirmed in adipocytes, muscle, liver, or β-cells. However, several studies have suggested the contribution of other G-proteins to the signaling of T2Rs.

One study showed that bitter-responsive taste receptor cells from α-gustducin-knockout mice retained 30% of their response to bitter stimuli [59]. Analysis of G-protein α-subunits showed that the inhibitory G-protein $G\alpha_{i2}$ was expressed in bitter-responsive cells, suggesting its coupling with T2Rs to sense bitter tastes [59]. Direct contact between T2Rs and inhibitory G-proteins was later shown by Sainz et al. via an in situ reconstitution assay [62]. Intestinal tuft cells were investigated for the contribution of G-protein subunits other than α-gustducin to T2R signaling, and immunohistochemical studies showed that $G\alpha_o$ is expressed in tuft cells [53]. $G\alpha_o$-specific inhibitor blocked the immune response induced by an extract of the parasitic helminth *Trichinella spiralis*, which is reported to activate T2Rs, thus suggesting the involvement of $G\alpha_o$ in T2R signaling [53]. The contribution of inhibitory G-proteins to the T2R-mediated relaxation of smooth muscle has also been reported. Kim et al. showed that human airway smooth muscle expresses α-gustducin and $G\alpha_o$ at the limit of detection level, whereas $G\alpha_i$ was found to be abundant [60]. They also showed the contribution of $G\alpha_i$ through a knockdown experiment, in which a bitter compound-stimulated increase in intracellular $Ca^{2+}$ concentration was not affected by α-gustducin and $G\alpha_o$ knockdown, but $G\alpha_{i1}$, $G\alpha_{i2}$, and $G\alpha_{i3}$ knockdown reduced the

response. The contribution of inhibitory G-protein to T2R signaling has also been shown in the T2R- and T1R-expressing model system of HEK293 cells [61]. These studies showed that G-proteins other than gustducin may participate in cells that lack α-gustducin, and signaling pathways such as cAMP signaling may contribute to the cellular effect of bitter compounds through T2Rs (Figure 2b).

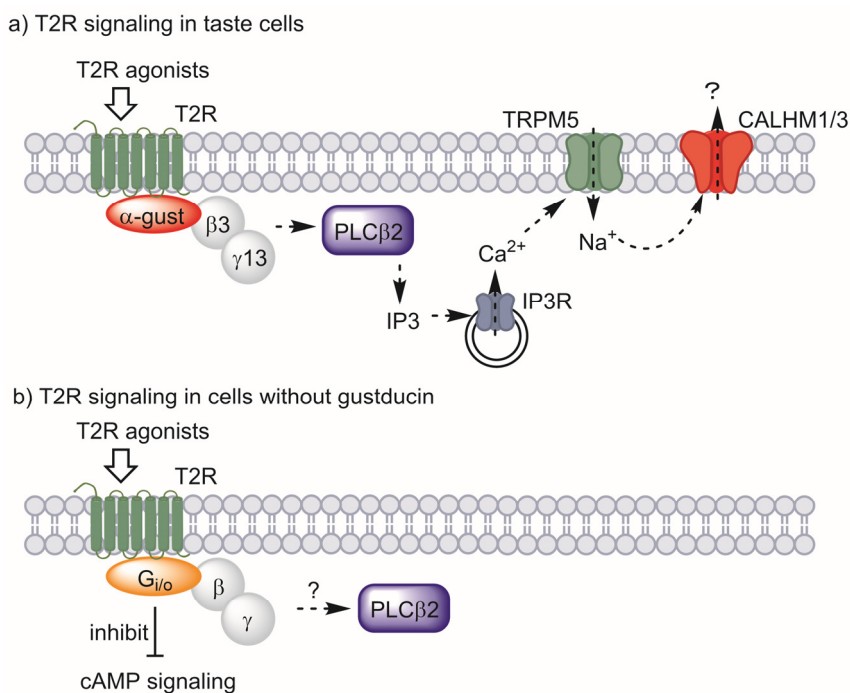

**Figure 2.** T2R signaling in taste cells (**a**) and proposed signaling for cells without gustducin (**b**). α-Gustducin (α-gust), beta-3 (β3), and gamma-13 (γ13) act as the major G-protein in taste signaling. Other G-proteins may have this role in other types of cells. Abbreviations: taste 2 receptor (T2R), α-gustducin (α-gust), beta-3 (β3), gamma-13 (γ13), phospholipase C β2 (PLCβ2), inositol 1,4,5-triphosphate (IP3), IP3 receptor (IP3R), transient receptor potential cation channel, subfamily M, member 5 (TRPM5), calcium homeostasis modulator 1/3 (CALHM1/3).

## 5. Conclusions

The findings presented in this paper suggest that promoting the secretion of the hormone incretin from enteroendocrine cells and regulating adipocyte differentiation may be the primary roles of T2Rs related to the development of obesity and diabetes. Given that T2Rs are also expressed in other tissues, those T2Rs may also be associated with obesity and diabetes. Many questions remain: Do T2Rs actually exist as functional proteins in each tissue? If T2Rs are functional, with which G-proteins do they associate? In addition, most importantly, what are the ligands for T2Rs in each tissue? Answering these questions will provide a deeper understanding of whether the T2Rs in each tissue work to regulate lipid/glucose metabolism, which would indicate their relationships with obesity/diabetes. Furthermore, the current knowledge is insufficient to reveal the link between T2Rs in each tissue and the development of diabetes or obesity. Further studies in animal models, including the use of T2R-knockout animals and promising and specific T2R agonists coupled with specific antagonists, are required to clearly show the role of extraoral T2Rs. Moreover, clinical trials are necessary to extrapolate the findings obtained thus far to humans.

**Author Contributions:** Writing—original draft preparation, E.K.; writing—review and editing, E.K. and S.O. All authors have read and agreed to the published version of the manuscript.

**Funding:** This research was funded by JSPS KAKENHI, grant numbers 23K05107.

**Acknowledgments:** The Genotype-Tissue Expression (GTEx) Project was supported by the Common Fund of the Office of the Director of the National Institutes of Health and by NCI, NHGRI, NHLBI, NIDA, NIMH, and NINDS. The data used for the analyses described in this manuscript were obtained from the GTEx portal on 21 July 2023.

**Conflicts of Interest:** The authors declare no conflict of interest.

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
