# Peer review of "Association of Bitter Taste Receptors with Obesity and Diabetes and Their Role in Related Tissues"

_2813-2564, doi:10.3390/receptors2040017_

Round 1
Reviewer 1 Report
Comments and Suggestions for Authors
This manuscript provides a review of the literature examining taste 2 receptors (T2Rs) in obesity and diabetes. The review presents a summary of current literature examining T2Rs in the context of obesity and diabetes. The authors argue that- 1) since T2Rs are found in extra-oral tissues that play a role in obesity and diabetes and that 2) since mutations in T2Rs associated tissues are observed in obesity and diabetes- T2Rs may be involved in the development of obesity and diabetes. While this is indeed a good hypothesis, it would be helpful if the authors could discuss other scientific theories to support this argument. This further explanation may strengthen this stance. The manuscript is well written, however- there are few grammatical and language edits that could be made to strengthen the manuscript. I strongly suggest that the authors include a table with a summary of the literature described in the review. The section on muscle is missing justification/background as to why T2R expression in muscle is relevant to diabetes and obesity. Additionally, a few clarification points and edits are recommended to improve the manuscript (listed below).
Line 13- Please clarify what “those” refers to in the sentence “the tissues those are associated with”.
Abstract (lines 10-18)- there are multiple places where “those tissues” is used. Does this refer to tissues and organs that are affected by or play a role in obesity in diabetes? If so, please clarify this and also list these tissues and organs. There are many tissues and organs affected by obesity and diabetes. If the hypothesis that T2R receptors are associated with diabetes and obesity is arising from the presence of T2R receptors in such tissues/organs, these tissues/organs must be clearly identified.
Line 24- “humans and rodents have approximately 25 and 35 functional T2R genes” Is it 25 in humans and 35 in rodents- if such either state or add the workd respectively at the end of the sentence.
Line 39- “suggestive” is incorrectly used. Should be “suggested role”.
Second paragraph/ starting on line 41- What type of review is this. This should be clearly identified and included in the introduction and abstract. Is it a narrative review? Scoping review? Systematic review?
Line 41-42- “and then discuss findings of” is grammatically incorrect. I would suggest “then, the findings of ___are discussed.”
Line 43- Why are the agonists that target T2Rs summarized in Table 1? What is the connection of these agonists to the previous sentence. A transition or connection between the previous sentence and this sentence is needed. \
Line 48- The location of the “,” and the “and” are incorrect. Perhaps “relation of T2R, obesity, and diabetes”?
Line 49- Are non-human T2Rs not involved in the regulation of body fat mass?
Line 51- The sentence starting with “studies have suggested an association”- is this referring to clinical studies (in people), pre-clinical, or both? Since this first paragraph appears to focus on clinical studies, this should be clarified. This should be clear throughout the manuscript since it discusses both clinical studies and pre-clinical.
Line 59- It is unclear what “works in adipose tissue” means. What is its role? Or is it just present?
Lines 69-72- How is it that altering ofloxacin, procainamide, etc. suggests that this increases diabetes risk? I know what each drug does but how does this relate to diabetes? This relationship should clearly be described, the reader should not have to figure it out on their own.
Lines 75-78- Please clarify if this work is conducted by the authors or another research team. If not conducted by the authors, the study team should be referenced in these sentences.
Line 83- Again clarify/state the name of the research team responsible for the data discussed in these sentences.
Lines 98-99- The sentence should be rephrased. Perhaps “Since T2Rs are expressed in enteroendocrine cells, it is likely that they are involved in obesity and diabetes.” The current order of the sentence makes it difficult for the reader to understand.
Line 99- This sentence should be written in the present tense. Therefore, this sentence should say “regulate” rather than regulated.
Line 102- Please clarify whether “that” refers to decreased secretion of GLP-1. The way it is written it is difficult to ascertain what “that” is referring to.
Line 122-126- This sentence refers to research conducted by the authors. If this work is published, it should be cited. If it is not, it should be clearly identified as a work in progress.
Lines 130-132- The sentence refers to “two studies” that have demonstrated the role of T2Rs in adipocyte differentiation. However, only one citation (citation 28) is provided. Please provide a citation for the second study. If the second study is referring to a study by the authors please clarify this.
Lines 132 and 134- Please clarify what “they” is referring to. Is it referring to the study 28 or the uncited second study?
Line 137-139- The sentence is difficult to understand. Perhaps it could be rephrased to “ Research by our group showed expression of …..in 3T3-L1 cells.”
Line 154- The sentence is missing a noun. “Two research [missing noun] suggest that….” Two research studies?
Lines 161- 170- clarify which of these studies are human vs. animal/mouse. The previous sections had each identified clinical and pre-clinical studies, but this section does not.
Section 3.3 Muscle- Why is the expression of T2R in muscle relevant to diabetes and obesity? This section needs justification/background to explain why T2R in muscle is relevant. For example, this section describes that T2Rs may be present in airway smooth muscle- but how is this relevant to diabetes? A clear link must be described by the authors so that the reader comprehends the relevance and importance of this section.
Lines 219-220- The last sentence in this section needs more detail (see above comment). Please add more detail to describe how muscle relaxation and contraction by T2R could prevent or treat diabetes. What is the link?
Section 3.4 Liver- Again, more detail is needed as to why the liver is important in diabetes and obesity.
Line 230- This section is labeled “3.4 Pancreatic islets”. However, the previous section was also 3.4. Is this section 3.4 or 3.5?
Section 4- More information is needed to explain why signaling pathways of T2Rs is important in diabetes and obesity. Perhaps just a few sentences would be needed to capture the “big picture.” Why does this matter? How does it relate to obesity and diabetes?
Figure 2- The figure gives a good summary of the transduction pathway. Are these pathways applicable to both diabetes and obesity? If so, how? Where is this transduction cascade occurring in the body? Taste cells in the tongue/taste buds? In animals and humans (mice/drosophila/mammals) or just humans? Which type of taste cells (e.g., Type I, Type II)? Does it occur in extra-oral tissues that also express T2Rs?
Overall- The discussion of how each section tells us about T2Rs and obesity and diabetes needs to be richer. Although some things might be obvious (the pancreas obviously plays a role in diabetes) the story should be well developed and defined in a review. The reader should not be left to speculate how different sections relate to the title/overall theme. It should be clearly described by the authors.The manuscript is well written and has good summaries, but is missing more information to capture the bigger picture that is claimed in the title.
Comments on the Quality of English LanguageThe manuscript is well written, however- there are few grammatical edits and language edits that could be made to strengthen the manuscript.
Author Response
Attached as a file

Reviewer 2 Report
Comments and Suggestions for Authors
This paper reviews the extra oral expressions and functions of bitter taste receptors, T2Rs, from the points of views in obesity and diabetes. While some points are necessary to be revised, it is systematically and compactly written.
Minor points:
Line 53
Three positions of single nucleotide polymorphism
L62
Signal from T2R as well as T1R
L63, L66
T2R and/or T1R
L81
Why the association with the development was suggested? Is it not enough only for the diet?
L94
In mice [20]
L138
Our group
Author Response
Attached as a file

Reviewer 3 Report
Comments and Suggestions for Authors
This review is pretty concise but informative, would be helpful to the researchers in the field, particularly to those who just started to work in this interesting area. But some minor revisions are needed:
1) Line 24: rats have 36 T2R genes. So it is somewhat misleading to state "... rodents have ...35 functional T2R genes."
2) Line 138. Since there are two authors, it is unclear whose group is "my group".
3) Line 154, "Two research suggest" seems quite unusual.
4) Lines 248 and 266: TRPM5 stands for transient receptor potential cation channel subfamily M, member 5. Not membrane 5.
Overall, it is a quite good manuscript.
Comments on the Quality of English LanguageSee above
Author Response
Attached as a file

Reviewer 4 Report
Comments and Suggestions for Authors
The paper would be greatly improved if edited by a company specializing in drafting scientific papers in English. Here are some obvious edits but more would be necessary (see also red-line pdf copy of the paper).
Author Response
Attached as a file

Round 2
Reviewer 4 Report
Comments and Suggestions for Authors
Thank you for the revised version. Much improved